# DIFFERENTIALLY PRIVATE FEDERATED LEARNING: A CLIENT LEVEL PERSPECTIVE

## ABSTRACT

Federated learning is a recent advance in privacy protection. In this context, a trusted curator aggregates parameters optimized in decentralized fashion by multiple clients. The resulting model is then distributed back to all clients, ultimately converging to a joint representative model without explicitly having to share the data. However, the protocol is vulnerable to differential attacks, which could originate from any party contributing during federated optimization. In such an attack, a client's contribution during training and information about their data set is revealed through analyzing the distributed model. We tackle this problem and propose an algorithm for client sided differential privacy preserving federated optimization. The aim is to hide clients' contributions during training, balancing the trade-off between privacy loss and model performance. Empirical studies suggest that given a sufficiently large number of participating clients, our proposed procedure can maintain client-level differential privacy at only a minor cost in model performance.

## 1 INTRODUCTION

Lately, the topic of security in machine learning is enjoying increased interest. This can be largely attributed to the success of big data in conjunction with deep learning and the urge for creating and processing ever larger data sets for data mining. However, with the emergence of more and more machine learning services becoming part of our daily lives, making use of our data, special measures must be taken to protect privacy. Unfortunately, anonymization alone often is not sufficient Narayanan & Shmatikov (2008); Backstrom et al. (2007) and standard machine learning approaches largely disregard privacy aspects and are susceptible to a variety of adversarial attacks Melis et al. (2018). In this regard, machine learning can be analyzed to recover private information about the participating user or employed data as well Oh et al. (2018); Shokri et al. (2017); Dang et al. (2017); Fredrikson et al. (2015). Carlini et al. (2018) propose a measure for to assess the memorization of privacy related data. All the aspects of privacy-preserving machine learning are aggravated when further restrictions apply such as a limited number of participating clients or restricted communication bandwidth such as mobile devices Google (2017).

In order to alleviate the need of explicitly sharing data for training machine learning models, decentralized approaches have been proposed, sometimes referred to as collaborative Shokri & Shmatikov (2015) or federated learning McMahan et al. (2017). In federated learning McMahan et al. (2017) a model is learned by multiple clients in decentralized fashion. Learning is shifted to the clients and only learned parameters are centralized by a trusted curator. This curator then distributes an aggregated model back to the clients. However, this alone is not sufficent to preserve privacy. In Orekondy et al. (2018) it is shown that clients be identified in a federated learning setting by the model updates alone, necessitating further steps.

Clients not revealing their data is an advance in privacy protection. However, when a model is learned in conventional way, its parameters reveal information about the data that was used during training. In order to solve this issue, the concept of differential privacy (dp) Dwork (2006) for learning algorithms was proposed by Abadi et al. (2016). The aim is to ensure a learned model does not reveal whether a certain data point was used during training.

We propose an algorithm that incorporates a dp-preserving mechanism into federated learning. However, opposed to Abadi et al. (2016) we do not aim at protecting w.r.t. a single data point only. Rather, we want to ensure that a learned model does not reveal whether a client participated during

decentralized training. This implies a client's whole data set is protected against differential attacks from other clients.

Our main contributions: First, we show that a client's participation can be hidden while model performance is kept high in federated learning. We demonstrate that our proposed algorithm can achieve client level differential privacy at a minor loss in model performance. An independent study McMahan et al. (2018), published at the same time, proposed a similar procedure for client level-dp. Experimental setups however differ and McMahan et al. (2018) also includes element-level privacy measures. Second, we propose to dynamically adapt the dp-preserving mechanism during decentralized training. Empirical studies suggest that model performance is increased that way. This stands in contrast to latest advances in centralized training with differential privacy, were such adaptation was not beneficial. We can link this discrepancy to the fact that, compared to centralized learning, gradients in federated learning exhibit different sensibilities to noise and batch size throughout the course of training.

## 2 BACKGROUND

### 2.1 FEDERATED LEARNING

In federated learning McMahan et al. (2017), communication between curator and clients might be limited (e.g. mobile phones) and/or vulnerable to interception. The challenge of federated optimization is to learn a model with minimal information overhead between clients and curator. In addition, clients' data might be non-IID, unbalanced and massively distributed. The algorithm 'federated averaging' recently proposed by McMahan et al. (2017), tackles these challenges. During multiple rounds of communication between curator and clients, a central model is trained. At each communication round, the curator distributes the current central model to a fraction of clients. The clients then perform local optimization. To minimize communication, clients might take several steps of mini-batch gradient descent during a single communication round. Next, the optimized models are sent back to the curator, who aggregates them (e.g. averaging) to allocate a new central model. Depending on the performance of the new central model, training is either stopped or a new communication round starts. In federated learning, clients never share data, only model parameters.

### 2.2 LEARNING WITH DIFFERENTIAL PRIVACY

A lot of research has been conducted in protecting differential privacy on data level when a model is learned in a centralized manner. This can be done by incorporating a dp-preserving randomized mechanism (e.g. the Gaussian mechanism) into the learning process.

We use the same definition for differential privacy in randomized mechanisms as Abadi et al. (2016):

A randomized mechanism $M : D \to R$, with domain $D$ and range $R$ satisfies $(\epsilon, \delta)$-differential privacy, if for any two adjacent inputs $d, d' \in D$ and for any subset of outputs $S \subseteq R$ it holds that $P[M(d) \in S] \le e^\epsilon Pr[M(d') \in S] + \delta$. In this definition, $\delta$ accounts for the probability that plain $\epsilon$-differential privacy is broken.

The Gaussian mechanism (GM) approximates a real valued function $f : D \to R$ with a differentially private mechanism. Specifically, a GM adds Gaussian noise calibrated to the functions data set sensitivity $S_f$. This sensitivity is defined as the maximum of the absolute distance $\|f(d) - f(d')\|_2$, where $d'$ and $d$ are two adjacent inputs. A GM is then defined as $M(d) = f(d) + \mathcal{N}(0, \sigma^2 S_f^2)$.

In the following we assume that $\sigma$ and $\epsilon$ are fixed and evaluate an inquiry to the GM about a single approximation of $f(d)$. We can then bound the probability that $\epsilon$-dp is broken according to: $\delta \le \frac{5}{4} \exp(-(\sigma\epsilon)^2/2)$ (Theorem 3.22 in Dwork & Roth (2014)). It should be noted that $\delta$ is accumulative and grows if the consecutive inquiries to the GM. Therefore, to protect privacy, an accountant keeps track of $\delta$. Once a certain threshold for $\delta$ is reached, the GM shall not answer any new inquires.

Recently, Abadi et al. (2016) proposed a differentially private stochastic gradient descent algorithm (dp-SGD). dp-SGD works similar to mini-batch gradient descent but the gradient averaging step is approximated by a GM. In addition, the mini-batches are allocated through random sampling of the data. For $\epsilon$ being fixed, a privacy accountant keeps track of $\delta$ and stops training once a threshold is

reached. Intuitively, this means training is stopped once the probability that the learned model reveals whether a certain data point is part of the training set exceeds a certain threshold.

### 2.3 CLIENT-SIDED DIFFERENTIAL PRIVACY IN FEDERATED OPTIMIZATION

We propose to incorporate a randomized mechanism into federated learning. However, opposed to Abadi et al. (2016) we do not aim at protecting a single data point's contribution in learning a model. Instead, we aim at protecting a whole client's data set. That is, we want to ensure that a learned model does not reveal whether a client participated during decentralized training while maintaining high model performance.

## 3 PROPOSED METHOD

In the framework of federated optimization McMahan et al. (2017), the central curator averages client models (i.e. weight matrices) after each communication round. In our proposed algorithm, we will alter and approximate this averaging with a randomized mechanism. This is done to hide a single client's contribution within the aggregation and thus within the entire decentralized learning procedure.

The randomized mechanism we use to approximate the average consists of :

- **Random sub-sampling** (step 1 in Fig. 1): Let $K$ be the total number of clients. In each communication round a random subset $Z_t$ of size $m_t \leq K$ is sampled. The curator then distributes the central model $w_t$ to only these clients. The central model is optimized by the clients' on their data. The clients in $Z_t$ now hold distinct local models $\{w^k\}_{k=0}^{m_t}$. The difference between the optimized local model and the central model will be referred to as client $k$'s update $\Delta w^k = w^k - w_t$. The updates are sent back to the central curator at the end of each communication round.

- **Distorting** (step 3 and 4 in Fig. 1): A Gaussian mechanism is used to distort the sum of all updates. This requires knowledge about the set's sensitivity with respect to the summing operation. We can enforce a certain sensitivity by using scaled versions instead of the true updates: $\triangle \bar{w}^k = \triangle w^k / \max(1, \frac{\|\triangle w^k\|_2}{S})$. Scaling ensures that the second norm is limited $\forall k, \|\triangle \bar{w}^k\|_2 < S$. The sensitivity of the scaled updates with respect to the summing operation is thus upper bounded by $S$. The GM now adds noise (scaled to sensitivity $S$) to the sum of all scaled updates. Dividing the GM's output by $m_t$ yields an approximation to the true average of all client's updates, while preventing leakage of crucial information about an individual.

A new central model $w_{t+1}$ is allocated by adding this approximation to the current central model $w_t$.

$$w_{t+1} = w_t + \frac{1}{m_t}(\overbrace{\sum_{k=0}^{m_t} \triangle w^k / \max(1, \frac{\|\triangle w^k\|_2}{S})}^{\text{Sum of updates clipped at } S} + \overbrace{\mathcal{N}(0, \sigma^2 S^2)}^{\text{Noise scaled to } S})$$
$$\underbrace{\phantom{\frac{1}{m_t}(\sum_{k=0}^{m_t} \triangle w^k / \max(1, \frac{\|\triangle w^k\|_2}{S}) + \mathcal{N}(0, \sigma^2 S^2))}}_{\text{Gaussian mechanism approximating sum of updates}}$$

When factorizing $1/m_t$ into the Gaussian mechanism, we notice that the average's distortion is governed by the noise variance $S^2 \sigma^2 / m$. However, this distortion should not exceed a certain limit. Otherwise too much information from the sub-sampled average is destroyed by the added noise and there will not be any learning progress. GM and random sub-sampling are both randomized mechanisms. (Indeed, Abadi et al. (2016) used exactly this kind of average approximation in dp-SGD. However, there it is used for gradient averaging, hiding a single data point's gradient at every iteration). Thus, $\sigma$ and $m$ also define the privacy loss incurred when the randomized mechanism provides an average approximation.

In order to keep track of this privacy loss, we make use of the moments accountant as proposed by Abadi et al. Abadi et al. (2016). This accounting method provides much tighter bounds on the incurred privacy loss than the standard composition theorem (3.14 in Dwork & Roth (2014)). Each time the curator allocates a new model, the accountant evaluates $\delta$ given $\epsilon$, $\sigma$ and $m$. Training shall be

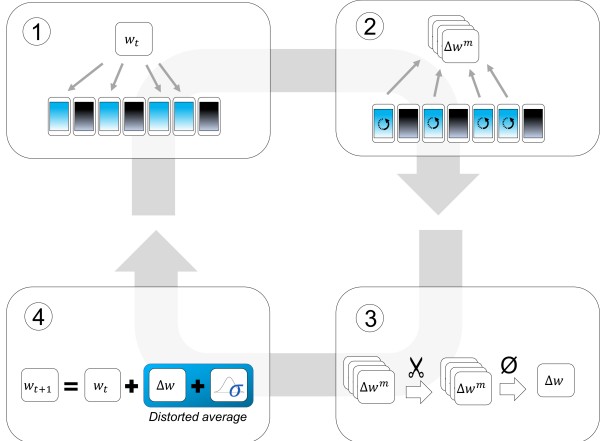

Figure 1: Illustration of differentially private federated learning on a client level. Step 1: At each communication round $t$, $m_t$ out of total $K$ clients are sampled uniformly at random. The central model $w_t$ is distributed to the sampled clients. Step 2: The selected clients optimize $w_t$ on their local data, leading to $w^k$. Clients centralize their local updates: $\triangle w^k = w^k - w_t$. Step 3: The updates are clipped such that their sensitivity can be upper bounded. The clipped updates are averaged. Step 4: The central model is updated adding the averaged, clipped updates and distorting them with Gaussian noise tuned to the sensitivity's upper bound. Having allocated the new central model, the procedure can be repeated. However, before starting step 1, a privacy accountant evaluates the privacy loss that would arise through performing another communication round. If that privacy loss is acceptable, a new round may start.

stopped once $\delta$ reaches a certain threshold, i.e. the likelihood, that a clients contribution is revealed gets too high. The choice of a threshold for $\delta$ depends on the total amount of clients $K$. To ascertain that privacy for many is not preserved at the expense of revealing total information about a few, we have to ensure that $\delta \ll \frac{1}{K}$, refer to Dwork & Roth (2014) chapter 2.3 for more details.

**Choosing $S$:** When clipping the contributions, there is a trade-off. On the one hand, $S$ should be chosen small such that the noise variance stays small. On the other hand, one wants to maintain as much of the original contributions as possible. Following a procedure proposed by Abadi et al. (2016), in each communication round we calculate the median norm of all unclipped contributions and use this as the clipping bound $S = \text{median}\{\triangle w^k\}_{k \in Z_t}$. We do not use a randomized mechanism for computing the median, which, strictly speaking, is a violation of privacy. However, the information leakage through the median is small (Future work will contain such a privacy measure).

**Choosing $\sigma$ and $m$:** for fixed $S$, the ratio $r = \sigma^2/m$ governs distortion and privacy loss. It follows that the higher $\sigma$ and the lower $m$, the higher the privacy loss. The privacy accountant tells us that for fixed $r = \sigma^2/m$, i.e. for the same level of distortion, privacy loss is smaller for $\sigma$ and $m$ both being small. An upper bound on the distortion rate $r$ and a lower bound on the number of sub-sampled clients $\bar{m}$ would thus lead to a choice of $\sigma$. A lower bound on $m$ is, however, hard to estimate. That is, because data in federated settings is non-IID and contributions from clients might be very distinct. We therefore define the between clients variance $V_c$ as a measure of similarity between clients' updates.

**Definition.** *Let $\triangle w_{i,j}$ define the $(i,j)$-th parameter in an update of the form $\triangle w \in \mathbb{R}^{q \times p}$, at some communication round $t$. For the sake of clarity, we will drop specific indexing of communication rounds for now.*

*The variance of parameter $(i,j)$ throughout all $K$ clients is defined as,*

$$VAR[\triangle w_{i,j}] = \frac{1}{K} \sum_{k=0}^{K} (\triangle w_{i,j}^k - \mu_{i,j})^2,$$

*where $\mu_{i,j} = \frac{1}{K} \sum_{k=1}^{K} \triangle w_{i,j}^k$.*

We then define $V_c$ as the sum over all parameter variances in the update matrix as,

$$V_c = \frac{1}{q \times p} \sum_{i=0}^{q} \sum_{j=0}^{p} VAR[\triangle w_{i,j}].$$

Further, the Update scale $U_s$ is defined as,

$$U_s = \frac{1}{q \times p} \sum_{i=0}^{q} \sum_{j=0}^{p} \mu_{i,j}^2.$$

---

**Algorithm 1** Client-side differentially private federated optimization. $K$ is the number of participating clients; $B$ is the local mini-batch size, $E$ the number of local epochs, $\eta$ is the learning rate, $\{\sigma\}_{t=0}^{T}$ is the set of variances for the GM. $\{m_t\}_{t=0}^{T}$ determines the number of participating clients at each communication round. $\epsilon$ defines the dp we aim for. $Q$ is the threshold for $\delta$, the probability that $\epsilon$-dp is broken. $T$ is the number of communication rounds after which $\delta$ surpasses $Q$. $\mathcal{B}$ is a set holding client's data sliced into batches of size B

---

1: **procedure** SERVER EXECUTION
2:     Initialize: $w_0$, Accountant$(\epsilon, K)$            ▷ initialize weights and the priv. accountant
3:     **for** each round $t = 1, 2, ...$ **do**
4:         $\delta \leftarrow$ Accountant$(m_t, \sigma_t)$        ▷ Accountant returns priv. loss for current round
5:         **if** $\delta > Q$ **then** return $w_t$       ▷ If privacy budget is spent, return current model
6:         $Z_t \leftarrow$ random set of $m_t$ clients       ▷ randomly allocate a set of $m_t$ out of K clients
7:         **for** each client $k \in Z_t$ in parallel **do**
8:             $\triangle w_{t+1}^k, \zeta^k \leftarrow$ ClientUpdate$(k, w_t)$     ▷ client k's update and the update's norm
9:         $S = \text{median}\{\zeta^k\}_{k \in Z_t}$           ▷ median of norms of clients' update
10:        $w_{t+1} \leftarrow w_t + \frac{1}{m}(\sum_{k=1}^{K} \triangle w_{t+1}^k / \max(1, \frac{\zeta^k}{S}) + \mathcal{N}(0, S^2 \cdot \sigma^2))$ ▷ Update central model
11: **function** CLIENTUPDATE$(k, w_t)$
12:     $w \leftarrow w_t$
13:     **for** each local Epoch $i = 1, 2, ...E$ **do**
14:         **for** batch $b \in \mathcal{B}$ **do**
15:             $w \leftarrow w - \eta \nabla L(w; b)$           ▷ mini batch gradient descent
16:     $\triangle w_{t+1} = w - w_t$           ▷ clients local model update
17:     $\zeta = \|\triangle w_{t+1}\|_2$               ▷ norm of update
18:     return $\triangle w_{t+1}, \zeta$          ▷ return clipped update and norm of update

---

## 4 EXPERIMENTS

In order to test our proposed algorithm we simulate a federated setting. For the sake of comparability, we choose a similar experimental setup as McMahan et al. (2017) did. We divide the sorted MNIST set into shards. Consequently, each client gets two shards. This way most clients will have samples from two digits only. A single client could thus never train a model on their data such that it reaches high classification accuracy for all ten digits.

We are investigating differential privacy in the federated setting for scenarios of $K \in \{100, 1000, 10000\}$. In each setting the clients get exactly 600 data points. For $K \in \{1000, 10000\}$, data points are repeated.

For all three scenarios $K \in \{100, 1000, 10000\}$ we performed a cross-validation grid search on the following parameters:

- Number of batches per client $B$
- Epochs to run on each client $E$
- Number of clients participating in each round $m$
- The GM parameter $\sigma$

Table 1: Experimental results for differentially private federated learning and comparison to non-differentially private federated learning (Non-DP). Scenarios of 100, 1000 and 10000 clients for a privacy budget of $\epsilon = 8$. $\delta'$ is the highest acceptable probability of $\epsilon$-differential privacy being broken. Once $\delta'$ is reached, training is stopped. Accuracy denoted as 'ACC', number of communication as 'CR' and communication costs as 'CC'.

| CLIENTS | $\delta'$ | ACC | CR | CC |
|---|---|---|---|---|
| 100 | NON-DP | 0.97 | 380 | 38000 |
| 100 | $e-3$ | 0.78 | 11 | 550 |
| 1000 | $e-5$ | 0.92 | 54 | 11880 |
| 10000 | $e-6$ | 0.96 | 412 | 209500 |

In accordance to Abadi et al. (2016) we fixed $\epsilon$ to the value of 8. During training we keep track of privacy loss using the privacy accountant. Training is stopped once $\delta$ reaches $e-3, e-5, e-6$ for 100, 1000 and 10000 clients, respectively. In addition, we also analyze the between clients variance over the course of training.

## 5 RESULTS

In the cross validation grid search we look for those models that reach the highest accuracy while staying below the respective bound on $\delta$. In addition, when multiple models reach the same accuracy, the one with fewer needed communication rounds is preferred.

Table 1 holds the best models found for $K \in \{100, 1000, 10000\}$. We list the accuracy (ACC), the number of communication rounds (CR) needed and the arising communication costs (CC). Communication costs are defined as the number of times a model gets send by a client over the course of training, i.e. $\sum_{t=0}^{T} m_t$. In addition, as a benchmark, Table 1 also holds the ACC, CR and CC of the best performing non-differentially private model for $K = 100$. In Fig. 2, the accuracy of all four best performing models is depicted over the course of training.

In Fig. 3, the accuracy of non-differentially private federated optimization for $K = 100$ is depicted again together with the between clients variance and the update scale over the course of training.

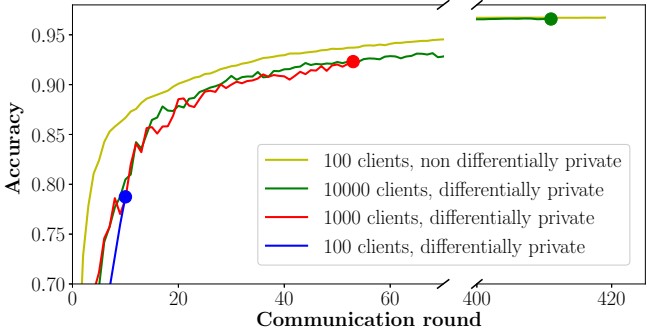

Figure 2: Accuracy of digit classification from non-IID MNIST-data held by clients over the course of decentralized training. For differentially private federated optimization, dots at the end of accuracy curves indicate that the $\delta$-threshold was reached and training therefore stopped.

## 6 DISCUSSION

As intuitively expected, the number of participating clients has a major impact on the achieved model performance. For 100 and 1000 clients, model accuracy does not converge and stays significantly below the non-differentially private performance. However, 78% and 92% accuracy for $K \in \{100, 1000\}$ are still substantially better than anything clients would be able to achieve when only

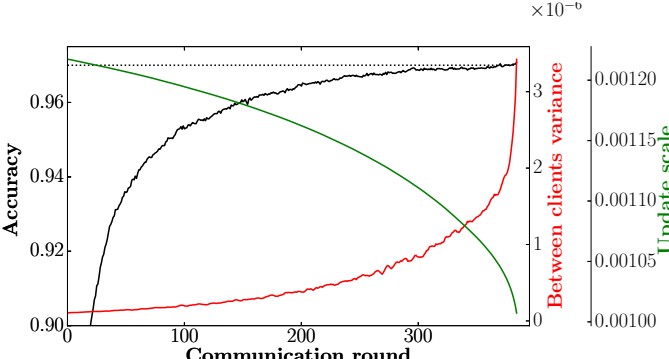

Figure 3: Scenario of 100 clients, non-differentially private: accuracy, between clients variance and update scale over the course of federated optimization.

training on their own data. In domains where $K$ lays in this order of magnitude and differential privacy is of utmost importance, such models would still substantially benefit any client participating. An example for such a domain are hospitals. Several hundred could jointly learn a model, while information about a specific hospital stays hidden. In addition, the jointly learned model could be used as an initialization for further client-side training.

For $K = 10000$, the differentially private model almost reaches accuracies of the non-differential private one. This suggests that for scenarios where many parties are involved, differential privacy comes at almost no cost in model performance. These scenarios include mobile phones and other consumer devices.

In the cross-validation grid search we also found that raising $m_t$ over the course of training improves model performance. When looking at a single early communication round, lowering both $m_t$ and $\sigma_t$ in a fashion such that $\sigma_t^2/m_t$ stays constant, has almost no impact on the accuracy gain during that round. however, privacy loss is reduced when both parameters are lowered. This means more communication rounds can be performed later on in training, before the privacy budget is drained. In subsequent communication rounds, a large $m_t$ is unavoidable to gain accuracy, and a higher privacy cost has to be embraced in order to improve the model.

This observation can be linked to recent advances of information theory in learning algorithms. As observable in Fig. 3, Shwartz-Ziv and Tishby Shwartz-Ziv & Tishby (2017) suggest, we can distinguish two different phases of training: label fitting and data fitting phase.

During label fitting phase, updates by clients are similar and thus $V_c$ is low, as Fig. 3 shows. $U_c$, however, is high during this initial phase, as big updates to the randomly initialized weights are performed. During data fitting phase $V_c$ rises. The individual updates $\triangle w^k$ look less alike, as each client optimizes on their data set. $U_c$ however drastically shrinks, as a local optima of the global model is approached, accuracy converges and the contributions cancel each other out to a certain extend. Figure 3 shows these dependencies of $V_c$ and $U_c$.

We can conclude: $i$) At early communication rounds, small subsets of clients might still contribute an average update $\triangle w_t$ representative of the true data distribution $ii$) At later stages a balanced (and therefore bigger) fraction of clients is needed to reach a certain representativity for an update. $iii$) High $U_c$ makes early updates less vulnerable to noise.

## 7 CONCLUSION

We were able to show through first empirical studies that differential privacy on a client level is feasible and high model accuracies can be reached when sufficiently many parties are involved. Furthermore, we showed that careful investigation of the data and update distribution can lead to optimized privacy budgeting. For future work, we plan to derive optimal bounds in terms of signal to noise ratio in dependence of communication round, data representativity and between-client variance as well as further investigate the connection to information theory. Additionally, we plan

to further investigate the dataset dependency of the bounds. For assessing further applicability in bandwith-limited settings, we plan to investigate the applicability of proposed approach in context of compressed gradients such as proposed by Lin et al. (2018).

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
