# OpenReview forum: "Differentially Private Federated Learning: A Client Level Perspective"
_ICLR.cc/2019/Conference_

### Official Review · AnonReviewer3 · 2018-10-14
**interesting direction but confusing presentation**

**Rating:** 4
**Confidence:** 4

**Review:**

The main claim the authors make is that providing privacy in learning should go beyond just privacy for individual records to providing privacy for data contributors which could be an entire hospital. Adding privacy by design to the machine learning pipe-line is an important topic. Unfortunately, the presentation of this paper makes it hard to follow.

Some of the issues in this paper are technical and easy to resolve, such as citation format (see below) or consistency of notation (see below). Another example is that although the method presented here is suitable only for gradient based learning this is not stated clearly. However, other issues are more fundamental:
1.	The main motivation for this work is providing privacy to a client which could be a hospital as opposed to providing privacy to a single record – why is that an important task? Moreover, there are standard ways to extend differential privacy from a single record to a set of r records (see dwork & Rote, 2014 Theorem 2.2), in what sense the method presented here different than these methods?
2.	Another issue with the hospitals motivation is that the results show that when the number of parties is 10,000 the accuracy is close to the baseline. However, there are only 5534 registered hospitals in the US in 2018 according to the American Hospital Association (AHA): https://www.aha.org/statistics/fast-facts-us-hospitals. Therefore, are the sizes used in the experiments reasonable?
3.	In the presentation of the methods, it is not clear what is novel and what was already done by Abadi et al., 2016
4.	The theoretical analysis of the algorithm is only implied and not stated clearly
5.	In reporting the experiment setup key pieces of information are missing which makes the experiment irreproducible. For example, what is the leaning algorithm used? If it is a neural network, what was its layout? What type of cross validation was used to tune parameters?
6.	In describing the experiment it says that “For K\in\{1000,10000} data points are repeated.” This could mean that a single client holds the same point multiple times or that multiple clients hold the same data point. Which one of them is correct? What are the implications of that on the results of the experiment?
7.	Since grid search is used to tune parameters, more information is leaking which is not compensated for by, for example, composition bounds
8.	The results of the experiments are not contrasted against prior art, for example the results of Abadi et al., 2016.

Additional comments
9.	The introduction is confusing since it uses the term “federated learning” as a privacy technology. However federated learning discusses the scenario where the data is distributed between several parties. It is not necessarily the case that there are also privacy concerns associated, in many cases the need for federated learning is due to performance constraints.
10.	In the abstract the term “differential attacks” is used – what does it mean?
11.	“An independent study McMahan et al. (2018), published at the same time”- since you refer to the work of McMahan et al before your paper was reviewed, it means that the work of McMahan et al came out earlier.
12.	In the section “Choosing $\sigma$ and $m$” it is stated that the higher \sigma and the lower m, the higher the privacy loss. Isn’t the privacy loss reduced when \sigma is larger? Moreover, since you divide the gradients by m_t then the sensitivity of each party is of the order of S/m and therefore it reduces as m gets larger, hence, the privacy loss is smaller when m is large.
13.	At the bottom of page 4 and top of page 5 you introduce variance related terms that are never used in the algorithm or any analysis (they are presented in Figure 3). The variance between clients can be a function of how the data is split between them. If, for example, each client represents a different demography then the variance may be larger from the beginning.
14.	In the experiments (Table 1), what does it mean for \delta^\prime to be e-3, e-5 or e-6? Is it 10^{-3}, 10^{-5} and 10^{-6}?
15.	The methods presented here apply only for gradient descent learning algorithms, but this is not stated clearly. For example, would the methods presented here apply for learning tree based models?
16.	The citations are used incorrectly, for example “sometimes referred to as collaborative Shokri & Shmatikov (2015)” should be “sometimes referred to as collaborative (Shokri & Shmatikov, 2015)”. This can be achieved by using \citep in latex. This problem appears in many places in the paper, including, for example, “we make use of the moments accountant as proposed by Abadi et al. Abadi et al. (2016).” Which should be “we make use of the moments accountant as proposed by Abadi et al. (2016).” In which case you should use only \cite and not quote the name in the .tex file.
17.	“We use the same deﬁnition for differential privacy in randomized mechanisms as Abadi et al. (2016):” – the definition of differential privacy is due to Dwork, McSherry, Nissim & Smith, 2006
18.	Notation is followed loosely which makes it harder to follow at parts. For example, you use “m_t” for the number of participants in time t but in some cases,  you use only m as in “Choosing $\sigma$ and $m$”.
19.	In algorithm 1 the function ClientUpdate receives two parameters however the first parameter is never used in this function.
20.	Figure 2: I think it would be easier to see the results if you use log-log plot
21.	Discussion: “For K=10000, the differrntially private model almost reaches accuracies of the non-differential private one.” – it is true that the model used in this experiment achieves an accuracy of 0.97 without DP and the reported number for K=10000 is 0.96 which is very close. However, the baseline accuracy of 0.97 is very low for MNIST.
22.	In the bibliography you have Brendan McMahan appearing both as Brendan McMahan and H. Brendan McMahan


It is possible that underneath that this work has some hidden jams, however, the presentation makes them hard to find.

---

### Official Review · AnonReviewer2 · 2018-11-02
**Well-motivated problem, but incremental improvement over previous work?**

**Rating:** 4
**Confidence:** 3

**Review:**

[Post-rebuttal update] No author response was provided to address the reviewer comments. In particular, the paper's contributions and novelty compared with previous work seem limited, and no author response was provided to address this concern. I've left my overall score for the paper unchanged.

[Summary] The authors propose a protocol for training a model over private user data in a federated setting. In contrast with previous approaches which tried to ensure that a model would not reveal too much about any individual data point, this paper aims to prevent leakage of information about any individual client. (There may be many data points associated with a single client.)

[Key Comments] The submission generally seems polished and well-written. However, I have the impression that it's largely an incremental improvement over recent work by McMahan et al. (2018).
* If the main improvement of this paper over previous work is the dynamic adaptation of weight updates discussed in Section 3, the experimental results in Table 1 should compare the performance of the protocol with vs. without these changes. Otherwise, I think it would be helpful for the authors to update the submission to clarify their contributions.
* Updating Algorithm 1 / Line 9 (computation of the median weight update norm) to avoid leaking sensitive information to the clients would also strengthen the submission.
* It would also be helpful if the authors could explicitly list their assumptions about which parties are trusted and which are not (see below).

[Details]
[Pro 1] The submission is generally well-written and polished. I found the beginning of Section 3 especially helpful, since it breaks down a complex algorithm into simple/understandable parts.

[Pro 2] The proposed algorithm tackles the challenging/well-motivated problem of improving federated machine learning with strong theoretical privacy guarantees.

[Pro 3] Section 6 has an interesting analysis of how the weight updates produced by clients change over the course of training. This section does a good job of setting up the intuition for the training setup used in the paper, where the number of clients used in each round is gradually increased over the course of training.

[Con 1] I had trouble understanding the precise threat model used in the paper, and I think it would be helpful if the authors could update their submission to explicitly list their assumptions in one place. It seems like the server is trusted while the clients are not. However, I was unsure whether the goal was to protect against a single honest-but-curious client or to protect against multiple (possibly colluding) clients.

[Con 2] During each round of communication, the protocol computes the median of a set of values, each one originating from a different client, and the output of this computation is used to perform weight updates which are sent back to the clients. The authors note that "we do not use a randomized mechanism for computing the median, which, strictly speaking, is a violation of privacy. However, the information leakage through the median is small (future work will contain such privacy measures)." I appreciate the authors' honesty and thoroughness in pointing out this limitation. However, it does make the submission feel like a work in progress rather than a finished paper, and I think that the submission would be a bit stronger if this issue was addressed.

[Con 3] Given the experimental results reported in Section 4, it's difficult for me to understand how much of an improvement the authors' proposed dynamic weight updates provide in practice. This concern could be addressed with the inclusion of additional details and baselines:
* Few details are provided about the model training setup, and the reported accuracy of the non-differentially private model is quite low (3% reported error rate on MNIST; it's straightforward to get 1% error or below with a modern convolutional neural network). The authors say they use a setup similar to previous work by McMahan et al. (2017), but it seems like that paper uses a model with a much lower error rate (less than 1% based on a cursory inspection), which makes direct comparisons difficult.
* The introduction argues that "dynamically adapting the dp-preserving mechanism during decentralized training" is a significant difference from previous work. The claim could be strengthened if the authors extended Table 1 (experimental results for differentially private federated learning) in order to demonstrate the effect of dynamic adaptation on model quality.

---

### Official Review · AnonReviewer1 · 2018-11-03
**Differentially private variant of the federated learning framework**

**Rating:** 4
**Confidence:** 4

**Review:**

The paper revisits the federated learning framework from McMahan in the context of differential privacy.  The general concern with the vanilla federated learning framework is that it is susceptible to differencing attacks. To that end, the paper proposes to make the each of the interaction in the server-side component of the gradient descent to be differentially private w.r.t. the client contributions. This is simply done by adding noise (appropriately scaled) to the gradient updates.

My main concern is that the paper just described differentially private SGD, in the language of federated learning. I could not find any novelty in the approach. Furthermore, just using the vanilla moment's accountant to track privacy depletion in the federated setting is not totally correct. The moment's accountant framework in Abadi et al. uses the "secrecy of the sample" property to boost the privacy guarantee in a particular iteration. However, in the federated setting, the boost via secrecy of the sample does not hold immediately. One requirement of the secrecy of the sample theorem is that the sampled client has to be hidden. However, in the federated setting, even if one does not know what information a client sends to the servery, one can always observe if the client is sending *any* information. For a detailed discussion on this issue see https://arxiv.org/abs/1808.06651 .

---

### Public Comment · (anonymous) · 2018-10-19
**What's the contribution of this paper?**

This work is similar paradigm to LEARNING DIFFERENTIALLY PRIVATE RECURRENT LANGUAGE MODELS. So, what's the
difference to previous work?

---

> ### Public Comment · (anonymous) · 2018-10-19
> **Opposite extreme cases**
>
> Thank you very much for your question.
> We do point out the similarity to LEARNING DIFFERENTIALLY PRIVATE RECURRENT LANGUAGE MODELS in the introduction. The research was conducted at the same time as ours but published at last year’s ICLR-conference, whereas we presented ours at a workshop.
>
> The reason why we now decided to aim for a conference-publication is that the two research projects aimed at opposite extreme cases and we want to motivate research in ours.
> LEARNING DIFFERENTIALLY PRIVATE RECURRENT LANGUAGE MODELS shows that with lots of clients, performance of language models can be maintained high while privacy is ensured. The work is centered around mobile phone users where hundreds of millions of clients are a realistic scenario.
> DIFFERENTIALLY PRIVATE FEDERATED LEARNING: A CLIENT LEVEL PERSPECTIVE aims at the other extreme. We were primarily interested in institutions such as hospitals jointly learning models. In these scenarios, the number of clients (e.g. hospitals) could be as low as a hundred. We want to motivate research of differentially private federated learning in this less commercial area and point out its potential for hospitals, laboratories and universities that have high privacy standards but could greatly benefit from one another (e.g. the authors of [Multi-Institutional Deep Learning Modeling Without Sharing Patient Data: A Feasibility Study on Brain Tumor Segmentation] state that the integration of differential privacy into their research would make it applicable to sensitive data.) In our research we focus on different phases of federated learning and how low numbers of participants influence these phases and the privacy loss during them.
>
> TLDR;
> We did not want to show: 'We can include privacy into already existing language models learned from millions of clients without drawbacks'
> But instead: 'Hospitals, labs or universities, that do not cooperate in learning models as of today, could greatly benefit from one another without revealing sensitive information.'

---

### Public Comment · (anonymous) · 2018-10-19
**Please give a more clear desribtion of the algorithm part.**

It remains unclear in the algorithm. Firstly, what's the purpose to introduce the parameter variance V?
Then, the author used \sigma_t as the noise scale,  but used \sigma  in the following updating.
Please give some comment on how to change the noise scale at each iterative step.

---

> ### Public Comment · (anonymous) · 2018-10-22
> **Missing _t**
>
> Thank you for the comment. Indeed this is a mistake. In line 10 of the algorithm, \sigma must be replaced with \sigma_t.
> The parameter variance is introduced when defining the between clients variance (it is just an in-between step to make that definition easier).
>
> In the discussion we explain:
> For a certain noise scale at iteration t: \frac{sigma^2_t}{m_t}, the privacy loss is smaller for both sigma_t and m_t being small. Now if the clients provided very similar updates we would therefore go for small sigma_t and small m_t. But if the clients provided very distinct updates, a communication round with a small m_t would not improve the model even if the overall noise scale didn't change. (remember: In federated learning client-data might be non-IID).
>
> We show that over the course of federated learning (for highly non-IID clients) the similarity of updates decreases (between clients variance increases) and it is therefore advantageous to start with a low m_t and keep increasing it during subsequent iteration rounds. If \sigma_t is held constant for all t that means the noise scale decreases over the course of training.
>
> The precise choices of \sigma_t and \m_t over the course of training highly depend on the  federated learning scenario (the privacy budget, the data, the amount of clients and how data is distributed among them). We therefore cannot give a general iterative rule in the algorithm but just provide a tendency to be followed when these parameters are to be chosen for a new setting.

---

### Public Comment · (anonymous) · 2018-10-26
**The scale of the S**

It is unclear about the scale of the clip bound S. Could you please
add some details about the scale of the S, due to the S
is a key factor to the final performance.

---

> ### Public Comment · (anonymous) · 2018-10-30
> **Choosing S**
>
> Thank you very much for your question.
> On page 4, in the section 'Choosing S' we provide information about the scale. As proposed by Abadi et al. (2016) we chose S to be the median of the second norms of the client contributions. We thereby ensure that the noise does not explode when a client provides very large updates but also do not trim too much of the true updates.

---

### Meta-Review · Area_Chair1 · 2018-12-15
**Needs significant justification of novelty**

**Confidence:** 5
**Recommendation:** Reject

**Metareview:**

Following the unanimous vote of the reviewers, this paper is not ready for publication at ICLR. The greatest concern was that the novelty beyond past work has not been sufficiently demonstrated.